# The Holocene retreat dynamics and stability of Petermann Glacier in northwest Greenland

Martin Jakobsson[1,2], Kelly A. Hogan [3], Larry A. Mayer[4], Alan Mix[5], Anne Jennings [6], Joe Stoner[5], Björn Eriksson[1,2], Kevin Jerram[4], Rezwan Mohammad[1,2], Christof Pearce [7], Brendan Reilly[5] & Christian Stranne[1,2]

Submarine glacial landforms in fjords are imprints of the dynamic behaviour of marine-terminating glaciers and are informative about their most recent retreat phase. Here we use detailed multibeam bathymetry to map glacial landforms in Petermann Fjord and Nares Strait, northwestern Greenland. A large grounding-zone wedge (GZW) demonstrates that Petermann Glacier stabilised at the fjord mouth for a considerable time, likely buttressed by an ice shelf. This stability was followed by successive backstepping of the ice margin down the GZW's retrograde backslope forming small retreat ridges to 680 m current depth (~730–800 m palaeodepth). Iceberg ploughmarks occurring somewhat deeper show that thick, grounded ice persisted to these water depths before final breakup occurred. The palaeodepth limit of the recessional moraines is consistent with final collapse driven by marine ice cliff instability (MICI) with retreat to the next stable position located underneath the present Petermann ice tongue, where the seafloor is unmapped.

[1] Department of Geological Sciences, Stockholm University, 106 91 Stockholm, Sweden. [2] Bolin Centre for Climate Research, Stockholm University, 106 91 Stockholm, Sweden. [3] British Antarctic Survey, Natural Environment Research Council, High Cross, Madingley Road, Cambridge CB3 0ET, UK. [4] Center for Coastal and Ocean Mapping, University of New Hampshire, Durham, NH 03824, USA. [5] College of Earth, Ocean, and Atmospheric Sciences, Oregon State University, Corvallis, OR 97331, USA. [6] Institute of Arctic and Alpine Research, University of Colorado, Boulder, CO 80309-0450, USA. [7] Department of Geoscience, Aarhus University, 8000 Aarhus, Denmark. Correspondence and requests for materials should be addressed to M.J. (email: martin.jakobsson@geo.su.se)

Petermann Glacier is one of Greenland's larger and faster flowing marine-terminating outlet glaciers (Fig. 1)[1]. It drains the northwestern sector of the Greenland Ice Sheet into Nares Strait losing mass primarily through submarine melting[2,3], but also via episodic calving events from a floating ice shelf, hereafter referred to as an ice tongue, extending in front of the grounding line. Two major calving events in 2010 and 2012 shortened the ice tongue by ~35 km, reducing its extent by nearly 40%[4–6] (Fig. 1). The ice tongue is currently ~42 km long and 20 km wide. Relatively warm ($T > 0\,°C$) and salty ($S > 34.6$) subsurface water of Atlantic origin has been observed to enter the Petermann Fjord from the north via Robeson Channel and Hall Basin after flowing through the Arctic Ocean[5]. This warmer water has contributed to thinning of the ice tongue from below making it more prone to calving[5].

Two early maps of the Petermann Ice Tongue[7,8] show its margin close to the 2010 location prior to the major calving event (Fig. 1b). These two 'snapshots' provide limited insight into the former dynamics of Petermann Glacier; a longer time series of its behaviour and well-mapped boundary conditions, e.g., fjord bathymetry and regional geology, are required to improve predictions of the dynamic behaviour of Petermann Glacier[5,9]. Until now, neither detailed bathymetry for the fjord[10], nor information on the long-term dynamics of Petermann Glacier, have been available[9]. This is in contrast to several other Greenland's large marine-terminating outlet glaciers, where analyses of the geometry and distribution of submarine glacial landforms has facilitated reconstructions of their deglacial ice dynamics[11,12]. These landforms provide vital information on former ice extents, ice-flow directions, stability points and several other parameters directly related to ice dynamics, including the presence or absence of water or deformable till at the ice-bed interface[13]. Landform records can be used to constrain numerical models of ice-sheet retreat as well as to test theoretical processes of ice-sheet instability[14–16].

During late summer 2015, the *Petermann 2015 Expedition* using the Icebreaker (IB) *Oden*, targeted Petermann Fjord to investigate the marine cryosphere, oceanography and geology in the fjord and the adjacent Nares Strait region (Fig. 1). The marine programme consisted of geophysical mapping, sediment coring and oceanographic station work resulting in the near-complete mapping of the fjord by multibeam echosounders on IB *Oden* and the small survey boat RV *Skidbladner* (used to map near the shores). Here we analyse the submarine glacial landform record utilising the geophysical mapping data gathered during the *Petermann 2015 Expedition*. We identify a suite of glacial landforms providing information on the behaviour of Petermann Glacier that highlights the critical role of geology for the development of stability points in the seafloor landscape, where the ice retreat paused during deglaciation.

## Results

### Description of the seafloor morphology and fjord setting. The mapping shows that Petermann Fjord has an overdeepened central thalweg with a maximum water depth of 1158 m ~10 km seaward from where the ice tongue margin was located in August 2015 (Fig. 1b). Petermann Fjord is separated from Hall Basin in Nares Strait by a prominent bathymetric sill that is generally shallower than 400 m, but with a 443 m deep passage ~2 km west of the midpoint of the fjord entrance (Fig. 1). Unless otherwise specified, references to the ice tongue margin are with respect to its 2015 extent.

The eastern side of Petermann Fjord consists of exposed carbonate and other sedimentary rocks of Palaeozoic age[17], forming remarkably steep (some with slopes >70°) walls both above and below sea level. Seaward of the ice tongue margin, two outlet glaciers that drain an extension of the Greenland Ice Sheet debouch into the eastern side of Petermann Fjord.

The western side of the fjord displays an ~5-km-wide submarine plateau extending from the shore. This plateau is characterised by a set of nearly flat-lying terraces at different depths, with the deepest terrace, deeper than 600 m, in the central part and the shallowest with depths around 300 m in the northern, outer fjord, forming a part of the sill. The terrace steps most likely reflect underlying bedrock bedding planes, a phenomenon also observed on the sub-aerially exposed part of the western fjord wall. The bedrock planes are also observed in the western side of the fjord, ~10 km offshore from the ice tongue margin (Fig. 1). Two glaciers drain into the western side of Petermann Fjord, descending steeply from the mountain walls north of the ice tongue margin (Fig. 1b). We refer to the northernmost of these two unnamed glaciers as the *Skidbladner Glacier* since we mapped its grounded margin and related glacial landforms with our survey vessel RV *Skidbladner*. Huge blocks, some >100 m wide and >20 m high, probably originating from mass wasting of the steep fjord walls (Supplementary Figs. 1, 2), are visible in the multibeam bathymetry scattered on the seafloor and along the shores.

The seafloor morphology markedly changes character outside of the sill separating Petermann Fjord from Hall Basin. Both Kennedy and Robeson channels have generally smooth sediment-covered seafloors (Fig. 1). In between the smoother seafloor of these two channels is a large area, ~14 km wide and 64 km long, dominated by a fractured rough seafloor stretching diagonally across from the shore of Ellesmere Island in the southwest to the shore of the northwestern point of Hall Land in the northeast (Fig. 1b). Three relatively flat-topped shoals exist in Hall Basin marked S1–S3 in Fig. 1b. A fourth prominent shoal marked S4 has a more peaked crest, is shallower than 200 m and is located at the northern end of the rough seafloor area north of Hall Basin (Fig. 1b). This feature marks the border with the smoother and deeper Robeson Channel with the sharp transition between seafloor morphologies tracing the Wegener Transform Fault between Greenland and Ellesmere Island[18].

### Description and interpretation of glacial landforms. The seafloor of Petermann Fjord is heavily streamlined along the fjord direction (Figs. 1 and 2a, b) and there are signs of bedrock sculpting, particularly on the bathymetric plateau along the western side of the fjord. The first derivative of bathymetry, slope, accentuates the streamlined terrain (Fig. 2b). Longer streamlines consist of ridge-groove landforms that can be followed for up to ~10 km. The inter-ridge spacing varies widely from <100 m to occasionally >500 m, and the ridge heights measured from the bottom of adjacent grooves range from a few metres to ~50 m.

Morphologically, the streamlined ridge-groove landforms in Petermann Fjord conform to descriptions of mega-scale glacial lineations (MSGL), used as an indicator of fast-flowing, grounded ice[19,20]. Some of the streamlined landforms begin at a positive up-sticking crag or bedrock high and continue like stretched out tails in the direction of the sill, consistent with the sub-glacially formed landform referred to as crag-and-tails[20] (Fig. 2a). Among the mapped streamlined landforms in Petermann Fjord, a few have no apparent beginning at a positive-relief crag or bedrock high and have lower length-to-width ratios than the ridges belonging to MSGLs (Fig. 2a). We classify these as drumlins, a glacial landform generated beneath fast-flowing grounded ice where available sediment permits[20,21].

Sets of streamlined landforms, dominantly crag-and-tails, also exist outside of the sill in Hall Basin (Figs. 2c–f and 3a–d). The

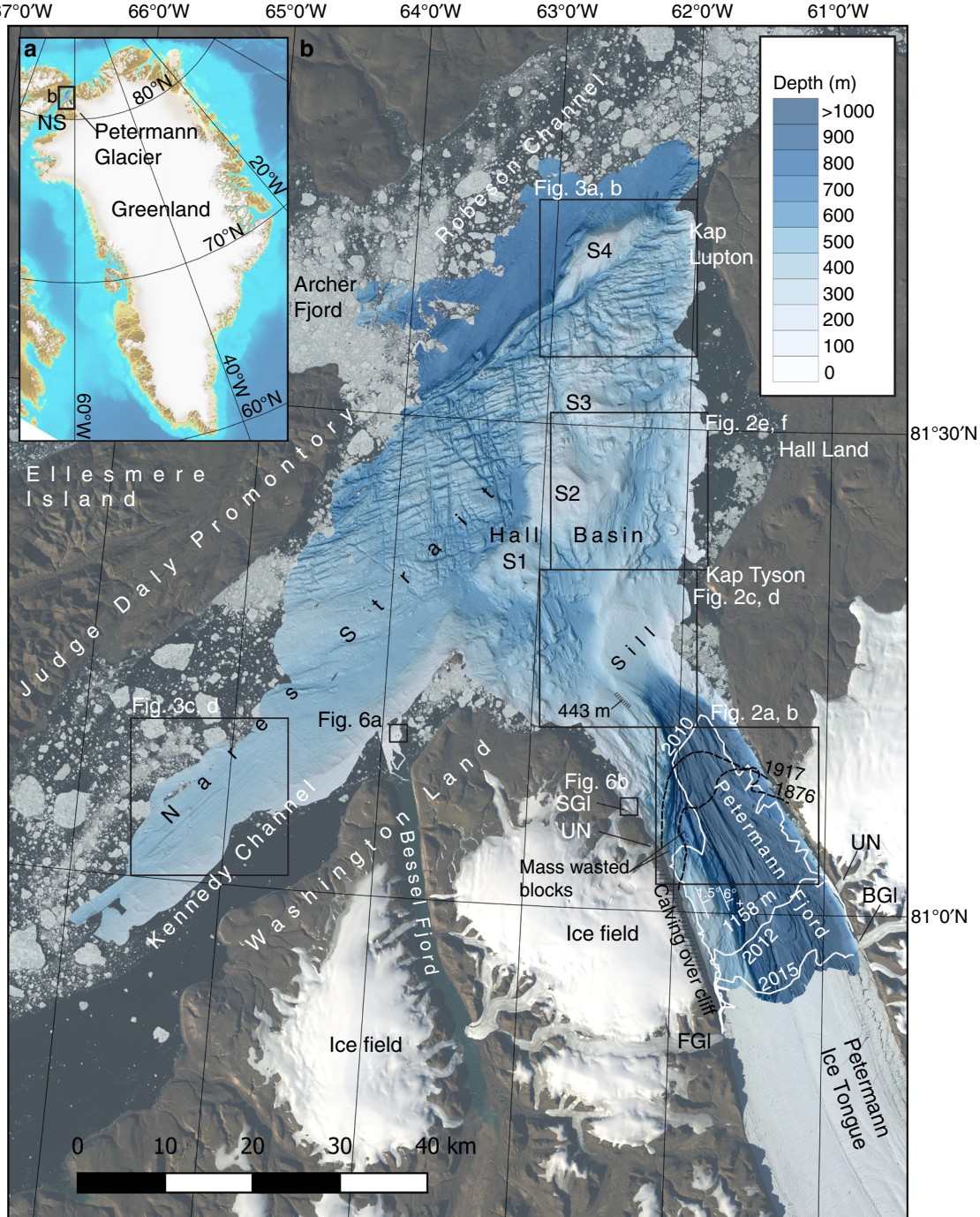

**Fig. 1** Overview maps of the studied area. **a** The location of Petermann Glacier and the focus area of the *Petermann 2015 Expedition* with Swedish icebreaker *Oden.* **b** The multibeam mapped area of the Petermann Fjord and adjacent Nares Strait. The bathymetry is portrayed with a shaded relief based on a 15 × 15 m cell-size grid from the processed multibeam bathymetry (see Methods). The location of the Petermann Ice Tongue margin is shown at five different points in time; for 2010 (July 22), 2012 (July 3) and 2015 (Aug 2), the margins are based on Landsat images (white lines), while the margins have been digitised from published maps for 1876 and 1917[7,8] (stippled black lines). Note that the 2010 and 2012 locations of the margin are prior to the main calving events that occurred those years. The areas of the detailed maps shown in Figs. 2–6 are outlined with black boxes. The strike and dip of the bedrock planes in the western side of the fjord as revealed by the multibeam bathymetry are marked as well as the deepest mapped location (1158 m) and the deepest spot in the sill (443 m). BGI Belgrade Glacier, FG1 Faith Glacier, NS Nares Strait, SGI Skidbladner Glacier, UN Un-Named glacier. S1–S4 are bathymetric shoals

orientations of these successively turn towards the north compared to those inside of the sill of Petermann Fjord. Finally, streamlined landforms are found both in Kennedy Channel and Robeson Channel, in the form of MSGLs, the former with higher length-to-width ratios and smaller relief compared to the MSGLs

in Petermann Fjord (Fig. 3a–d). In Robeson Channel, a subtle fluting, i.e., smaller and less pronounced than MSGLs, is seen, most visibly in the slope map (Fig. 3b). The shoal S4 is also fluted in the same direction as landforms in the nearby but deeper Robeson Channel. The streamlined landforms in both Kennedy

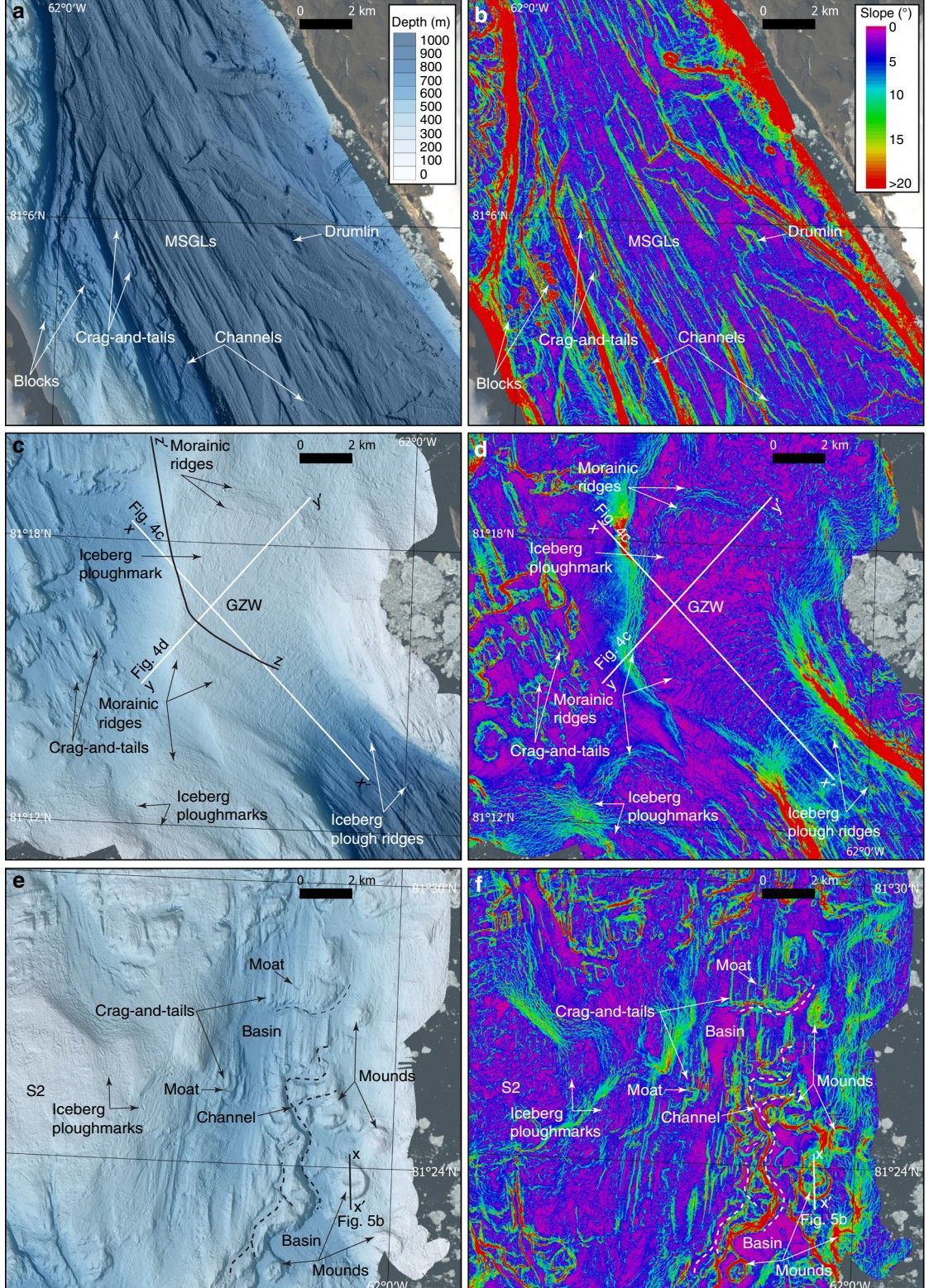

**Fig. 2** Detailed maps of the multibeam bathymetry and its first derivative, slope. **a**, **b** Petermann Fjord. **c**, **d** The fjord entrance and its bathymetric sill. GZW grounding-zone wedge. x–x′ and y–y′ show the locations of the bathymetric profiles in Fig. 4c, d. z–z′ in **c** shows the location of the sub-bottom profile in Supplementary Figure 3. **e**, **f** Eastern Hall Basin. The bathymetric profile between x and x′ is shown in Fig. 5b

Channel and Robeson Channel are parallel to the axes of the channels.

Directional analysis of all mapped streamlined landforms shows that within Petermann Fjord, i.e., inside the sill, the ice stream of Petermann Glacier followed the shape of the fjord (towards ~332°) and flowed through a 4-km-wide bottleneck before crossing the sill. In Hall Basin, the landforms show that ice flow progressively turned towards the northeast and eventually to a northerly direction from about 81°20′N (Fig. 2e, f). The ice stream that produced the streamlined landforms in Kennedy

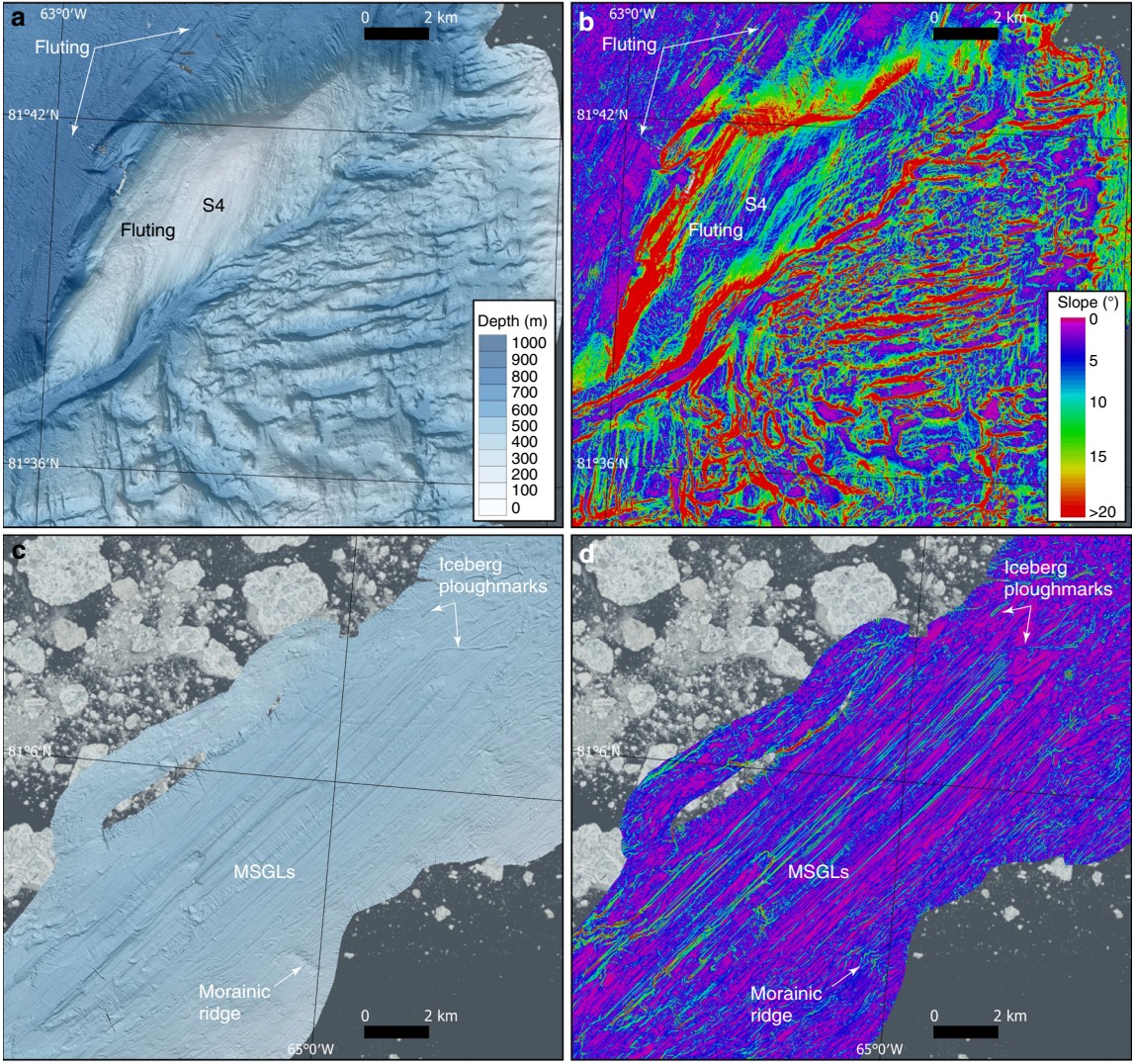

**Fig. 3** Detailed maps of the multibeam bathymetry and its first derivative, slope. **a**, **b** Northern part of Hall Basin and the adjacent Robeson Channel. **c**, **d** Kennedy channel. MSGL mega-scale glacial lineations

Channel followed the channel (towards ∼40°); in Robeson Channel, just northwest of the shoal S4, subtle fluting suggests ice flow towards ∼15°.

The sill separating Petermann Fjord from Hall Basin can morphologically be separated into northern and southern parts along a line representing the deepest water passage across the sill (Figs. 1b and 2c). The northern part has the form of a wedge with the steep front facing the Hall Basin (Figs. 2c and 4a). Its morphology and location suggest that it is a grounding-zone wedge (GZW), a constructional glacial landform formed at the grounded margins of marine-terminating glaciers, most likely in the presence of an ice shelf[22], from persistent sediment accumulation during halts in ice retreat[20,23] (Figs. 2c and 4a, b). The GZW at the northern sill of Petermann Fjord has a widest width of ∼8 km and a length, measured from its steep distal front to where it merges with the deep fjord trough, of ∼12 km (Fig. 4c, d). The GZW rises as much as 225 m from the base of the 4–9° steep front-facing Hall Basin (Fig. 4c). The streamlined landforms from Petermann Fjord reveal that the ice flow converged from the deep trough towards the trough bottleneck and GZW, and then spread out as it crossed the sill (Fig. 2c). At the final stage, when the GZW was fully developed and the grounding line was located on the sill, the ice stream must

have flowed upslope on its 2–4° sloping ice-proximal side before reaching the crest (Fig. 4c). We propose that shallow bedrock flanks of the fjord, just inside of the sill, acted to stabilise the grounding line there during retreat both by pinning on the shallow (<300 m depth) areas and because the trough narrows at that point providing lateral buttressing. Ice pinned on the southern flank of the sill was likely slower moving compared to the ice-stream flow focused through the deepest part of the trough leading to less sediment supply and a locally smaller and more diffuse GZW on the southern sill (Fig. 2c). The four shoals in Hall Basin (S1–S4) do not have the same wedge-like morphology as the GZW forming the northern sill of Petermann Fjord (Figs. 1b, 2e and 3a).

The surface of the GZW is characterised by small sinuous ridges with only a few metres of bathymetric expression (Figs. 2c and 4a). These ridges are particularly well portrayed in seafloor slope maps (Figs. 2d and 4b) and on a sub-bottom profile across the ridge from south to north (Supplementary Fig. 3). Quasi-parallel ridges trace the GZW crest in a regular pattern down its ice-proximal slope. We interpret these to be small morainic retreat ridges formed at the grounded tidewater margin of Petermann Glacier once it finally retreated from the fjord entrance. The implication of the recessional moraines, which

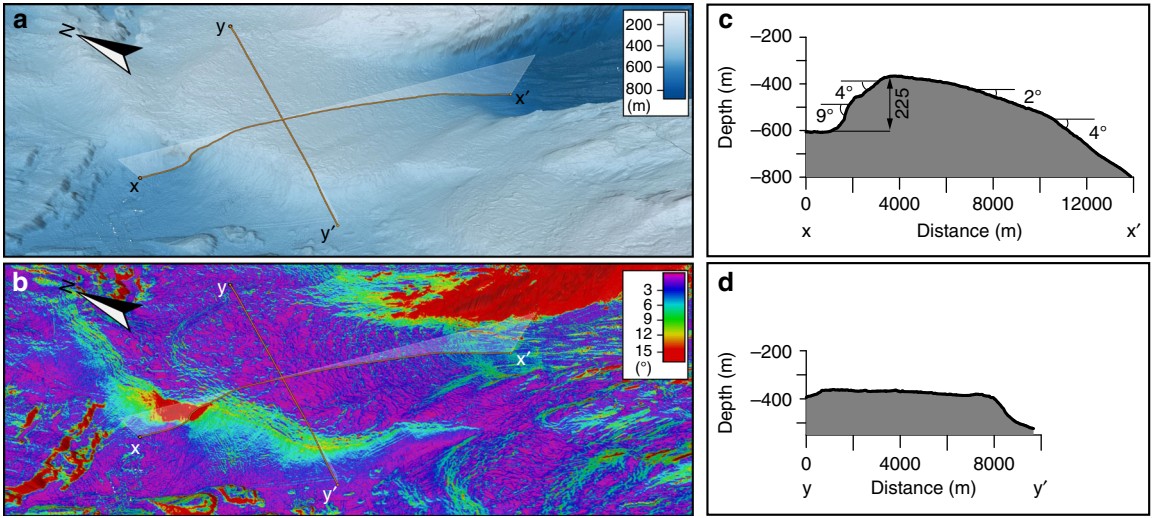

**Fig. 4** Bathymetric views of the entrance to Petermann Fjord, where the GZW is located. **a** Bathymetric 3D-view. **b** 3D-view of the same area as shown in **a**, but with calculated slope draped on top of the bathymetry. **c** Bathymetric profile between x and x'. Profile location is shown in **a** and **b**. **d** Bathymetric profile between y and y'. Profile location is shown in **a** and **b**

form at tidewater glacier margins (i.e., no ice tongue[24,25]) is that, if present, any ice tongue that buttressed Petermann Glacier during its halt at the fjord mouth broke up prior to its retreat from the sill. The pattern of the ridges suggests a regular backstepping of the glacier margin, which makes us suggest that there was no ice tongue in place when they were formed.

Larger ridges with lengths 1–1.5 km and heights 5–10 m are mapped in Kennedy Channel transverse to the ice-flow direction spaced by about 500 to >2500 m (Figs. 1b and 3c). These ridges are also sinuous but, in general, do not appear to extend across the channel. We suggest that these are also recessional moraines, although they are significantly more widely spaced than the ridges on the GZW.

The eastern part of Hall Basin displays prominent channels directly north of the GZW (Fig. 2e). Here, channels >200-m-wide wind through the seafloor over a >10-km-long area (Fig. 2e). Some channels appear to either start from or terminate at flat local basins (Fig. 2e). Arcuate moats also occur in this area on the ice-proximal side of crag-and-tails and extend approximately normal to the former ice-flow direction (Fig. 2e).

The morphology of the channels and their location resemble several of those found in Pine Island Bay, West Antarctica, where channels connecting local basins are interpreted to represent a well-developed subglacial drainage system[26,27], an interpretation we adopt. Moats located on the ice-proximal side of MSGLs or crag-and-tails are commonly prescribed a formation from erosion of subglacial meltwater[28], implying an abundance of subglacial meltwater in the eastern Hall Basin north of the GZW (Fig. 2e).

Six circular to semi-circular mounds are identified in the seafloor landscape in the eastern Hall Basin, north of the GZW (Figs. 2e, f and 5a, b). The largest three of these mounds rise as much as 100 m from the surrounding seafloor, have outer diameters of ~1.5 km and central depressions that are several hundred metres wide and up to 50 m deep. The depressions open out towards the side facing downslope and away from the coast, which makes them resemble giant horseshoes (Fig. 5a). The sides of the mounds slope in the range of 10–20°, although some sections have slopes >30°. Possible mechanisms for formation of the mounds are addressed in the discussion.

Throughout the region, bathymetric shoals and shallow areas along the coasts are heavily marked by iceberg ploughmarks (Figs. 2, 3 and 6). Areas shallower than ~400 m are particularly

heavily ploughed. The majority of the ploughmarks are a few metres deep and extend for some hundreds of metres, although larger ploughmarks stretching for several kilometres and with bathymetric expression of more than 10 m do exist. The ploughmarks on the GZW are at first difficult to separate from the retreat ridges, however, the systematic organisation of the latter following a retreating ice margin approximately parallel to the GZW crest makes it possible (Fig. 2c, d). At the start of the 2015 expedition, an ~1.5 × 3 km large iceberg calved off the ice tongue of Petermann Glacier and broke into several pieces that grounded on the shallow bank north of Bessel Fjord (Fig. 1 and Supplementary Fig. 4). We mapped the ploughmarks produced from one of the pieces (size: 200 × 400 m) that grounded in 68 m water depth (Fig. 6a). The relatively wide, but thin, iceberg had multiple keels that produced ploughmarks that were more than 2 m deep.

Where the deep trough of Petermann Fjord meets the GZW, the area of large MSGLs ends with furrows terminating in several distinct ploughed up ridges (Fig. 2c). The largest of these features has a groove that is ~300 m wide with sediment ridges ~4 m high. Identical landforms are described from the glacial trough of Pine Island Bay, where they were interpreted to have been formed when deep-drafting icebergs grounded on the up-sloping trough and rotated or further broke into pieces before floating away[29]. The iceberg grounding produced a furrow ending in a ploughed up ridge, hence the landform was referred to as an iceberg plough ridge[29]. Small, but clear versions of these iceberg plough ridges are mapped in front of the outlet glacier we refer to as Skidbladner Glacier (Fig. 6b and Supplementary Fig. 5).

## Discussion

The dynamics of glaciers, and particularly marine-terminating glaciers, are influenced by many factors in addition to climate, including ocean-ice interactions[30,31], bedrock controls[32], buttressing by floating ice shelves and fjord geometry[33].

The glacial landform record preserved on the seafloor will, under normal circumstances, be biased towards the final ice-retreat phase and breakup as earlier events are often erased by later events[20,34]. We find no evidence suggesting otherwise in Petermann Fjord or the adjacent Hall Basin and Robeson and Kennedy Channels of Nares Strait. This implies that the glacial landforms we mapped provide information on the retreat and

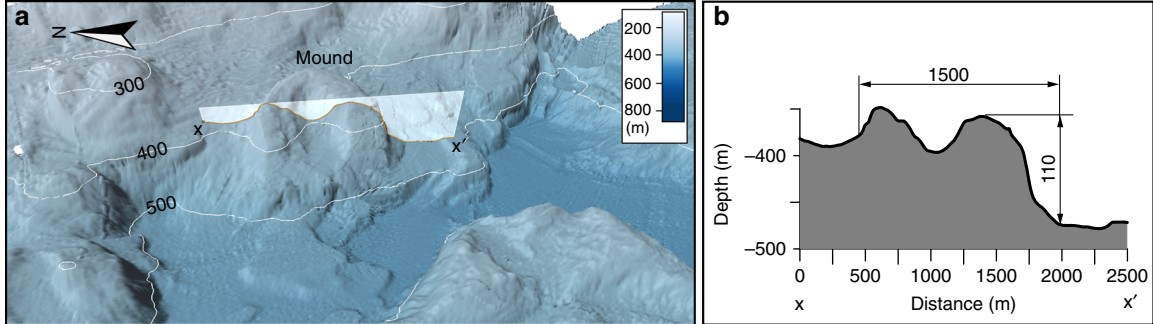

**Fig. 5** Bathymetric views of one of the horseshoe-shaped mounds. **a** Bathymetric 3D-view. This mound is located in the southeastern part of the detailed map in Fig. 2. **b** Bathymetric profile between x and x′ shown in **a**

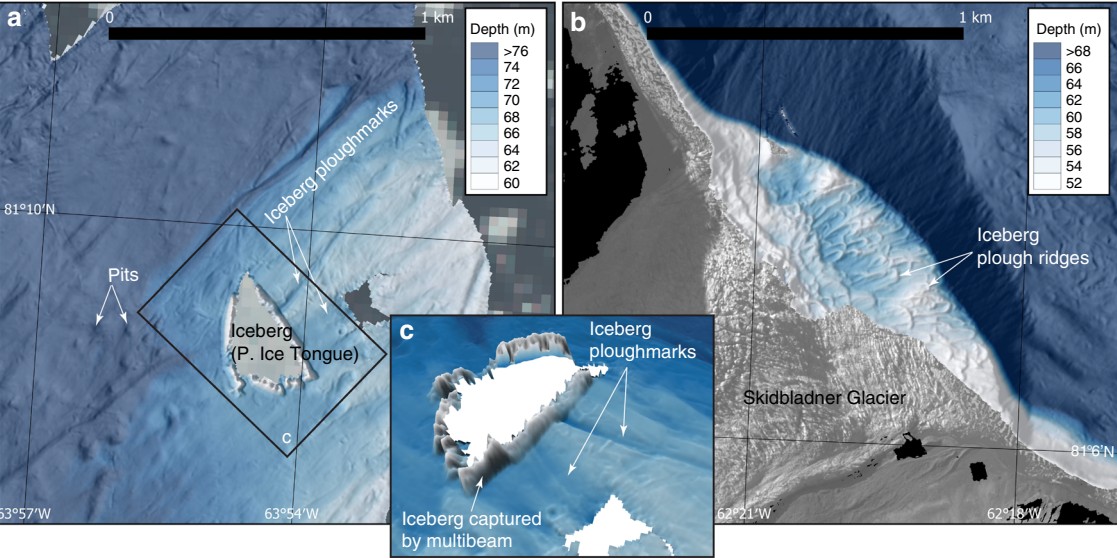

**Fig. 6** High-resolution bathymetry from the RV *Skidbladner* surveys. **a** Seafloor bathymetry of the area where one of the icebergs grounded that calved from the Petermann Ice Tongue during the expedition. **b** Seafloor bathymetry showing iceberg plough ridges in front of Skidbladner Glacier. **c** 3D-view of the mapped seafloor around the grounded iceberg. The sidewalls of the iceberg are mapped by the multibeam sonar and clearly visible in the data. The bathymetric shaded reliefs in **a**, **b** and **c** are based on processed grids (3 × 3 m cell size) from the RV *Skidbladner* surveys

breakup following the last glacial maximum (LGM), though it is possible that some glacial landforms (e.g., the large GZW at the Petermann Fjord entrance), may have been built over long time periods, even from multiple glaciations.

Erratic boulders from Greenland, documented as far as 20 km inland from the present north-eastern coast of Ellesmere Island, have been interpreted to show that the LGM Greenland Ice Sheet extended across Nares Strait coalescing with the Innuitian Ice Sheet on Ellesmere Island[35,36]. During the deglaciation, an ice stream confined to Nares Strait flowed north-eastward off Washington Land and Hall Land on the Greenland side[36]. Our mapping indicates that an ice stream debouched from Petermann Fjord and turned eastward to merge with an ice stream flowing towards the northeast in Nares Strait (Fig. 7). This fits the deglaciation pattern in Nares Strait presented by England[36] for 10,000, 9000 and 8500 radiocarbon years before present (BP), except that we do not find morphological evidence of ice turning west out of Petermann Fjord's southern side as suggested by England[36] (Fig. 7). The ages of the ice retreat margins by England[36] will henceforth be provided in calendar years BP. Calibrations have been made on the ice retreat margins using the Marine13 radiocarbon age calibration curve[37] and a $\Delta R = 268 \pm 82$ years, based on the five nearest analysed shells from Nares Strait and northern Baffin Bay[38]. England[36] applied a reservoir

correction of 410 years, which we accounted for when calibrating the radiocarbon years of the three ice margins presented in Fig. 7. Furthermore, considering the uncertainty in assigning a representative $\Delta R$ for the Petermann Fjord area, calibrations of the ice margin ages using $\Delta R = 0$ years are in addition presented in the caption of Fig. 7. These may be seen as maximum age constraints of the ice margins. Our mapping suggests that the ice stream emanating from Petermann Fjord (Petermann Ice Stream) was likely turned north-eastwards by ice already flowing through Nares Strait (Fig. 7). We cannot, however, constrain the presence or absence of such westward ice-flow pattern at an earlier stage. The fluted seafloor in Robeson Channel suggests that grounded ice continued to flow north-eastward beyond the shoal S4. The slight northward deflection of the lineations in Robeson Channel just beyond S4 may be a result of ice flow in Nares Strait merging with the Petermann Ice Stream.

At about 9300 cal. years BP, the ice margin had retreated to the area of Kap Lupton, northwestern Hall Land, from where it extended across the strait to the northeast Judge Daly Promontory (Fig. 7)[36]. Bedrock in the northernmost part of Hall Land comprises Late Silurian–Early Devonian limestones[39]. The seafloor morphology hints that the same type of bedrock extends across Hall Basin to the north-eastern tip of Judge Daly Promontory (Fig. 1). Since ice streams have a tendency to find

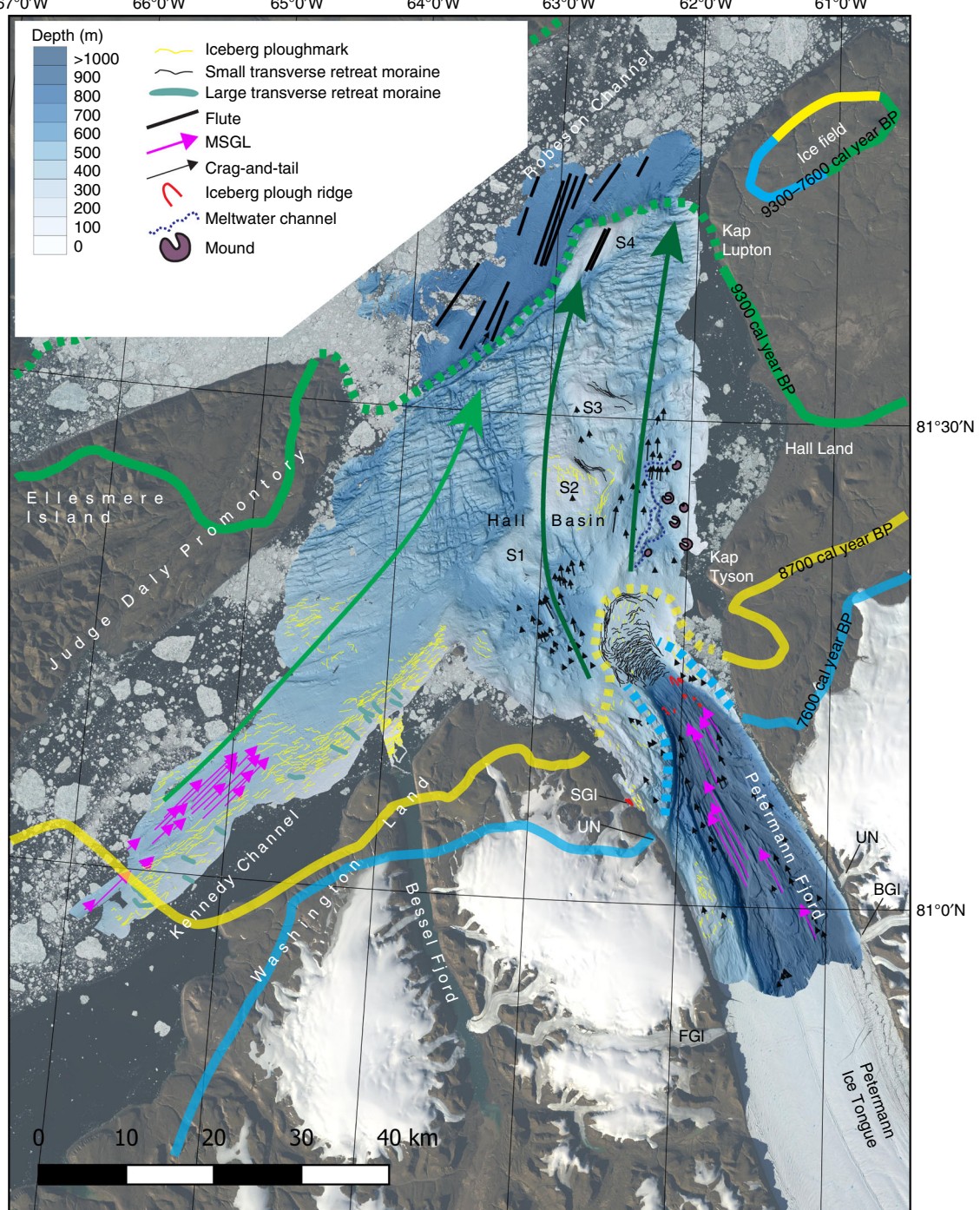

**Fig. 7** Interpretation of mapped glacial landforms in Petermann Fjord and adjacent areas of Hall Basin, Kennedy Channel and Robeson Channel. The ice retreat margins from England (1999) are shown for 9300 (green), 8700 (yellow) and 7600 (blue) cal. years BP. The ice margins are calibrated using $\Delta R = 268 \pm 82$ years. Applying $\Delta R = 0$ yields maximum calibrated ages of 9600, 9000 and 7900 cal. years BP. See main text for calendar year calibration of the C14 years (8500, 8000 and 7000) originally presented by England (1999). The stippled marine continuations of the ice margins are inferred in this work based on the mapped glacial landforms

stability points at natural bottlenecks or where resistant bedrocks make shoals, we suggest that the ice margin between Judge Daly Promontory and the Hall Land was grounded at 9300 cal. years BP along the apparent fault zone, where the bedrock and seafloor depth dramatically changes character (Fig. 7). We have no data that indicate whether an ice shelf extended from this grounding line over from Robeson Channel towards the Arctic Ocean at this time.

The abundant indications of meltwater next to the large mounds, some of which are horseshoe shaped (Fig. 2e), are here interpreted to indicate the formation of lateral shear margin that separated the fast-flowing, grounded Petermann Ice Stream from slower flowing, or perhaps even stagnant, ice on the shallower bank off western Hall Land (Fig. 7). Stagnant or slow-flowing ice commonly bounds fast-flowing ice streams, and the heavily crevassed shear zones in between the two act as a conduits for

meltwater; they also constitute the location where ice stream shear margin moraines form[40]. The ridge-like feature winding along the longest meltwater channel (Fig. 2e) may comprise a shear margin moraine or an esker-like feature following the underlying bedrock. In either case, the mounds are located in the area where the ice became stagnant or slow flowing and where meltwater was plentiful, inviting comparison with known glacial landforms. The lack of information about the geological composition of the mounds makes their origin difficult to determine. However, we have dismissed a submarine volcanic origin for the mounds as there are no signs of volcanic formations or activity in the nearby area. Considering that the mounds are located next to mapped carbonate platforms of the lower Palaeozoic Washington Land Group formation on Kap Tyson[39,41], we cannot exclude that they may be carbonate mounds[42]. Against this interpretation is their well-preserved appearance, considering that they have been overridden by the Petermann Glacier, and that central depressions not are common features of carbonate mounds. Another possibility is that the mounds comprise a glacial landform. They may represent kames, ridges or flat-topped mounds for which the formation is attributed to stagnant ice and meltwater[43,44]. Specific interpretations of how kames are formed vary greatly, although they generally are inferred to contain sediments initially accumulated primarily by meltwater on or in the ice, commonly in depressions or crevasses. The sediments are deposited on the bed after the glacier melts, often in ridges, mounds or terraces. It is not, however, obvious how the sediment would find its way through stagnant ice in a marine environment to form a nearly round, or horseshoe-shaped, kame on the seafloor in several hundred metres water depth. Stagnant marine ice would likely lift from the bed and disintegrate into icebergs before sediments on its surface, or englacial, melt through. It is especially hard to envision sediment deposition in a near-circular mound, occasionally with a central depression that is open towards one side, considering the ice breakup that would occur in a marine environment, which is distinctly different from a stagnant ice that slowly melts in a terrestrial environment.

Near-circular sediment mounds with central depressions do exist in the heavily glaciated landscape of North Dakota, USA[45]. These are referred to as 'doughnuts' and are interpreted to be so called 'dead-ice moraines'[45] formed when sediments fill a depression in stagnant ice and the ice eventually melts, leaving a mound of sediment with a core of ice inside. When the ice core eventually melts, the topography is inverted resulting in a central depression in the mound; many of these features also have an opening on the downstream end. Again, the marine environment poses problems for this explanation (i.e., keeping ice in contact with the seafloor) though enough englacial material could prevent the ice from floating.

Given the challenges of calling upon terrestrial analogues formed in stagnant ice for the formation of near-circular mounds in a marine environment, we find that we cannot readily adopt any previously published formation mechanism for similarly looking mounds, e.g., kames[44]. However, we suggest a mechanism where sediment accumulates on stagnant ice in depressions, much like that commonly proposed for kames. The accumulated sediment must eventually make its way through the ice by meltwater-aided transport, likely through systems of moulins or crevasses, to be deposited as a confined, high-relief mound on the seafloor once the stagnant ice breaks up and disintegrates. The horseshoe shape of some of the mounds remains an enigma. However, we speculate that it may originate from meltwater that breaches the edge of the deposited sediment mound once the central depression is formed, and flows towards the deeper side of the seafloor.

Field work on Hall Land suggests that the ice margin had retreated to the area south of Kap Tyson at 8700 cal. years BP[36] (Fig. 7). In marine areas, the exact timing of ice retreat and the corresponding opening of Nares Strait as an Arctic-Atlantic connection is not well known[46]. Yet the location of Kap Tyson adjacent to the GZW strongly suggests that the Petermann Ice Stream margin was located here at this time (Fig. 7).

The dimensions of the surface expression of the GZW are comparable with other larger GZWs, such as the largest in Pine Island Trough (GZW5), West Antarctica[47]. Considering sediment flux rates between 500 and 1650 $m^3 a^{-1} m^{-1}$, Jakobsson, et al.[47] estimated that the formation of GZW5 took between 600 and 2000 years. Estimating the volume of the Petermann GZW from multibeam sonar data and using the same flux rates as for the Pine Island Bay GZW5, in line with other West Greenland outlet glaciers[48], suggests that this feature may have accumulated in ~750–2480 years (Supplementary Figs. 6, 7). The ice margin reconstructions by England[36] suggest that the Petermann Ice Stream likely occupied the location of the GZW for about 1100 years, within our estimate using a GZW volume based on its surficial geometry. Even if the GZW accumulated over several glaciations, it likely represents a considerable stability point for the Petermann Ice Stream during its last retreat from Hall Basin into Petermann Fjord.

Current understanding of GZW formation implies the presence of a floating ice shelf beyond the grounding line, allowing the ice shelf to stabilise as the wedge grows vertically despite concurrent ice-sheet thinning[22,49]. We propose that Petermann Ice Stream was bounded by an ice shelf when it was pinned at the fjord mouth sill, with the ice shelf providing additional stability in the form of frontal buttressing[50]. Given evidence that Robeson Channel experienced reduced sea-ice cover from 9000 to 6000 cal. years BP[46] it is likely that the ice shelf did not extend that far out of the fjord (>30 km). Relatively warm Atlantic water ($T$, $S$) was present in Nares Strait by 9000 cal. years BP[46] and may have promoted ice shelf thinning to the point of ice shelf breakup, thus removing frontal buttressing and promoting retreat of Petermann Ice Stream into the fjord as a tidewater glacier margin.

The morainic ridges on the surface of the GZW indicate that retreat from the fjord mouth began with backstepping of the (now tidewater) ice-stream margin (Figs. 2c,b, 4a, b and 7) perhaps around 7600 cal. years BP as inferred from the location of the ice margin on land[36]. The ice-proximal (landward) side of the GZW has a retrograding slope between about 2° and 4°, implying that the ice stream margin flowed upslope during this retreat phase when the transverse morainic ridges were formed (Fig. 4c), presumably retaining stability from the constriction of the ice stream into the narrow, central trough just landward of the sill. The deepest mapped transverse morainic ridges are located at a present water depth of about 680 m with iceberg plough ridges occurring a bit deeper (Fig. 7). We find no morphological evidence from inner Petermann Fjord, inside of the GZW and moraines, suggesting that the Petermann Ice Stream made another halt during its retreat. In other words, following the formation of the deepest transverse moraines on the proximal retrograde slope of the GZW, the Petermann Ice Stream rapidly retreated throughout the remainder of the fjord we mapped (Fig. 1). Future mapping beneath the ice tongue may reveal additional grounding lines that are not visible at present. Of particular interest is a bedrock sill ~25 km from the modern grounding line with estimated water depth of 540–610 m[51], which could have provided an advanced stable position following retreat from the fjord mouth sill.

Weertman[52] suggested that marine ice streams flowing on retrograde slopes are inherently unstable, specifically when

lacking a buttressing ice tongue or shelf. If the vertical ice cliff at the margin rises more than about 100 m above sea level, the margin may begin fracturing and retreating rapidly, via massive calving, in a phenomena referred to as marine ice cliff instability (MICI)[53]. Assuming a density of ocean of water of 1026 kg m$^{-3}$ and of ice 917 kg m$^{-3}$, the draft of a floating ice mass is $\sim$0.89 $\times$ total ice thickness. This says that an ice cliff rising 100 m above sea level may form in the marine environment when the ice is grounded at water depths $\sim$809 m. Thus, grounding zones in more than 800 m water depth are considered to be particularly sensitive to MICI if there are no buttressing ice tongues or shelves in place[53]. Iceberg ploughmarks mapped in Pine Island Bay, West Antarctica, indicate that, during the last deglaciation, MICI was likely responsible for a rapid and sustained retreat of the Pine Island Glacier beginning around 12,300 cal. years BP[14]. The present depth of the Petermann Fjord (>1000 m), the retrograde landward slope of the GZW, and the location of iceberg plough ridges confirming calving all the way to the ice-stream base, lead us to suggest that MICI may have caused the Petermann Ice Stream to retreat rapidly from the fjord mouth sometime after 7600 cal. years BP. For $\sim$1100 years before the rapid retreat, the ice stream was stable at the fjord mouth facilitated by frontal buttressing by an ice shelf, lateral pinning points and a significant narrowing of the ice-stream trough. The palaeodepth of the GZW and Petermann Fjord can be derived for a given time if the isostatic depression of the area is known. At 7000 cal. years BP, the isostatic depression has been estimated to $\sim$70 m[54–56], which implies that the point where MICI would occur is merely 50 m deeper than the deepest mapped morainic ridge on the retrograde landward slope of the GZW.

Taken together, the subglacial landform record in Petermann Fjord and adjacent Hall Basin reveals the importance of geometric controls from the landscape, bedrock geology and of glaciological processes during the Holocene retreat of the Petermann Glacier to its present position, where the grounding line is located $\sim$90 km from the major GZW at the fjord entrance. The landform record shows that the fjord geometry likely played a role in determining where the Petermann Glacier was stable for perhaps as long as 1100 years, while glaciological processes in the form of MICI may have put a drastic end to this stable period, but only after the buttressing ice tongue broke up. Crucially, it was the interplay of all these factors that governed the final retreat of Petermann Glacier. Therefore, this study highlights the need to improve our knowledge of the whole glacier system, in particular high-resolution bathymetry in front of marine-terminating glaciers or beneath their ice tongues, before they can be accurately portrayed in ice-sheet models.

## Methods

**Geophysical mapping with IB Oden.** The *Petermann 2015 Expedition* started and ended in Thule, Greenland on July 30 and September 3, respectively. Systematic multibeam bathymetric mapping and sub-bottom profiling was carried out with IB *Oden* in Petermann Fjord and the adjacent areas of Nares Strait, including Hall Basin, Kennedy Channel and Robeson Channel (Fig. 1a, b). In total, $\sim$3100 km$^2$ was mapped with essentially full coverage from IB *Oden*, excluding the transit lines to and from the survey area. IB *Oden* has a hull-mounted Kongsberg EM 122 (12 kHz, 1° × 1°) multibeam echosounder and an SBP 120 (2–7 kHz, 3° × 3°) chirp sonar sub-bottom profiler installed. Navigation was provided by a Seatex Seapath 330 GPS/GLONASS navigation system integrating heading and attitude, and an Seapath MRU5 inertial motion sensor providing pitch, roll, yaw and heave. The accuracy of the positions from the combined GPS/GLONASS solution varied with available satellites, but was commonly reported to be within a 5 m sphere, or better, both vertically and horizontally. Regular CTD (Conductivity, Temperature, Depth) stations and XBTs (Expendable Bathy Thermograph) were used to provide the multibeam system on IB *Oden* with regular sound speed profiles of the water column.

**Geophysical mapping with RV Skidbladner.** In some near-shore areas, $\sim$30 km$^2$ of the seafloor was mapped from the 6.4-m-long RV *Skidbladner* using a Kongsberg EM 2040 (200–400 kHz, 1° × 1°) multibeam system bow mounted on a pole that can be hoisted up for transit. A Valeport sound velocity probe was used to acquire sound speed profiles. The navigation system of RV *Skidbladner* consists of a Seatex Seapath 330 + with RTK (Real Time Kinematic) capability. The motion sensor unit measuring roll, pitch, yaw and heave was a Seatex MRU5 + mounted in a subsea bottle on the EM 2040 transducer casing. During mapping operations, a Hemisphere A325 Smart antenna connected to a SATELLINE-EASy Pro 35 W 403–473 MHz radio was stationed on nearby topographic high points to provide RTK corrections sent using the RTCM 3.0 protocol. The accuracy of the GPS/GLONASS positions after RTK corrections was generally reported by the Seapath 330 + system to be within a 10 cm sphere, or better, both vertically and horizontally.

**Acquisition and processing software.** All multibeam data were acquired using Kongsberg Seafloor Information System (SIS) and processed using a combination of Caris and QPS Fledermaus processing software. The processed data were gridded to a horizontal resolution of 15 × 15 m for data from IB *Oden* and 3 × 3 m for data from RV *Skidbladner*. Seafloor morphology was interpreted in the 3D environment of Fledermaus and maps were subsequently produced using the Open Source GIS software QGIS (http://www.qgis.org). The chirp sub-bottom profiler data were acquired using the Kongberg TOPAS software and the profile displayed in this work was post-processed using tools provided by the Geological Survey of Canada (courtesy Bob Courtney).

**Data availability.** The multibeam bathymetry shown in this work is available for download from https://bolin.su.se/data/contributions.php?d=1609&p=MjAxOC0wNS0xOCAxNTowNDozOS4yNzg4NjMgMzk3ODknky8

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

## Acknowledgements

We thank US-NSF Polar Programs, NASA, UNAVCO, CH2M Polar Field Services (Jessy Jenkins), Polar Geospatial Center, the Swedish Polar Research Secretariat and the Swedish Maritime Administration for supporting the *Petermann 2015 Expedition*. K.A.H. was supported by the British Antarctic Survey 'Polar Science for Planet Earth' programme funded by the UK's Natural Environment Research Council. M.J. and colleagues from Stockholm University were supported by a grant from the Swedish Research Council (VR).

## Author contributions

M.J., K.A.H. and L.A.M. developed the concept of this paper and led the writing. A.M., L. A.M. and M.J. initiated the project and led the field work. A.M., A.J., J.S., K.J., C.P., B.R. and C.S. took part in the writing process and interpretation of results. B.E. and R.M. participated in the field work, data processing and evaluation.

## Additional information

**Competing interests:** The authors declare no competing interests.

