## [Peer Review File · Nature Communications]

Reviewers' comments:

Reviewer #1 (Remarks to the Author):

This paper uses exclusive new multibeam bathymetry data to map and interpret submarine glacial landforms in Petermann Fjord and Nares Strait, and to discuss the nature of the retreat of this significant glacier since the last glacial maximum. Of particular note, and therefore interest to the glaciological community, are the grounding zone wedge at the mouth of the fjord, which exhibits similarities to GZWs elsewhere in morphology and likely formation, and the potential in the measured bathymetry that the 'marine ice cliff instability' was responsible for the Holocene retreat of Petermann Glacier from the GZW.

The paper is well written and easy to follow, with extensive reference to the relevant literature, and the dataset it analyses appears to be unique and valuable. The nature of the paper is rather long, heavy on description and in places very speculative. Whilst it appears to fit the stated aims and scope of the journal, I was surprised to see it here rather than in a longer-format journal such as JGR. Many concepts and processes are discussed, and I was left wondering whether this would not have been better divided into two papers – a general descriptive presentation of the data and overall glaciological insights; and a short paper focussing on the two significant issues identified above.

Nevertheless, I believe the paper worthy of publication, but have the following concerns and suggestions which need to be addressed or explained.

COMMENTS (in the order I noted them in the paper)

- 1) The manuscript relies heavily on the figures, which I chose to review on paper. They are generally well designed, but most of the annotation is too small and difficult to read. Double the size of fonts to get twice the engagement and communication.
- 2) Dates are given sometimes in calendar years and sometime in radiocarbon years, and reference is made to calibration more than once. This is not my area, but I feel that simplifying this situation would make the paper easier to understand.
- 3) Likewise with water depths. Only in the Abstract is the discrepancy between present-day and palaeo water depths mentioned, yet this is crucial to some of the arguments. It should be made absolutely clear when depths are mentioned whether or not they are present day, and what impact the difference may have. Ideally, an explanation of the likely change in sea level since the formation of these features should be given, and water depths should be stated against the palaeo reference where that is relevant (e.g. MICI).
- 4) The pseudo-3D figures do not present well on a 2D medium and are very difficult to understand. I recommend on paper using only plan-view figures.
- 5) The allusion to a "giant toilet seat" (a) shouldn't have double-quotes (or probably any quotes at all) and (b) rather undermines the serious nature of the science. Use 'sliced donut' or something.
- 6) In Figure 6, it appears that the iceberg prevented bathymetric acquisition resulting in edge effects in the shaded representation. This needs fixing with some kind of mask.
- 7) Whilst possible, it is by no means certain (at least from the evidence presented) that the iceberg plough marks indicated were made by this iceberg, or even recently.
- 8) There is frequent reference to "stagnant (i.e. dead) ice". The "i.e. dead" adds nothing to this – was ice ever "alive"?
- 9) The discussion of the origin of the donut features is so speculative as to undermine the whole paper, so I recommend you just state that the origin is unknown. Unless there is previous record of "giant moulins" or moulins "shaped like a horseshoe", then the level of speculation is far too high.
- 10) The discussion of 'MICI' is confusing and needs more clarification. At one point the fjord depth (1000m) is used to invoke this argument, but line 446 seems to undermine this. These depths need better explanation with regard to palaeo sea level and isostasy. This is quite a big and trendy claim, so clarity on values is important.

11) The concluding paragraph is short and weak from such a lot of evidence. For people who tend to focus on the conclusions and figures, this paper would make little impression.

Reviewer #2 (Remarks to the Author):

This is a well written manuscript, although the detailed description of the morphological features is rather lengthy compared to the interpretation section and implication.

On the other hand, this is the first detailed seafloor map of this section and it is an important contribution to understanding deglaciation in this part of Greenland. Some morphological features, especially the proposed formation of mounts resembling "toilet seats" through moulins is very interesting and innovative.

Detailed comments:

Lines 138-143: This introduction to this section could be skipped. That figures 4 and 5 are 3D views and figure 6 is based on RV Skidbladner is already stated in the figure captions.

Line 209 "small sinuous ridges". I can't really see them in fig 4a only in 4b.

Line 215/216: recessional moraine have been reported from tide water glaciers, but it is not clear to me why they couldn't be formed by glaciers with an ice shelf. Tide-related movement of the grounding line has been reported for ice streams with ice selves in Antarctica (e.g. Bindshaler et al 2003 (10.1126/science.1087231), although we don't know, if they produce such ridges.

Line 333: "We have no morphological data". I suggest dropping "morphological since there are probably no other data, e.g. from sediments, either. Ice shelf presence is usually inferred from sediments rather from seafloor morphology.

Line 412 switch "Figs 2cb" to "Figs 2 b, c"

Figures

Figure 1:

- I suggest adding a label for Nare Strait on 1b (There is a label NS in 1a, but is not explained in captions)
- while the figures are very well drafted, the coloring of the labels (e.g. white on light blue) or black on dark blue is sometimes hard to see.
- maybe mark the older ice tongue locations as dashed white lines instead of black lines
- consider changing labels S1-S4 to darker color, e.g. black.

Figure 2

I am not sure the slope maps 2b, d, f, are necessary. In most cases the glacial features are clearer in the bathymetry maps with hill-shading. I suggest moving the slope maps to the supplement material.

Where the bathymetry is shown in light blue (2c, e) the labels would be easier to read in a darker color.

Figure 3:

Similar to Fig 2 I am not sure the slope maps 3b, d are necessary. In most cases the glacial features are clearer in the bathymetry maps with hill-shading. I suggest moving the slope maps to the supplement material.

Where the bathymetry is shown in light blue (3c) the labels would be easier to read in a darker color.

Figure 4:

Add a view direction to 3D view in Fig 4a and b

Figure 5:

Add a view direction to 3D view in Fig 4a and orientation to 5b

Figure 7:

There is yellow-blue-green circle in the top right corner. I don't think this marks an ice position but might have been a legend for the ice extent markers without labels. I suggest removing it or and labels.

It would be good to add the color indicator after the years in line 300. E.g. "8700 (green), 8150 (yellow), 7200 (blue)".

Answers to the Reviewers Comments

Reviewer #1 (Remarks to the Author):

This paper uses exclusive new multibeam bathymetry data to map and interpret submarine glacial landforms in Petermann Fjord and Nares Strait, and to discuss the nature of the retreat of this significant glacier since the last glacial maximum. Of particular note, and therefore interest to the glaciological community, are the grounding zone wedge at the mouth of the fjord, which exhibits similarities to GZWs elsewhere in morphology and likely formation, and the potential in the measured bathymetry that the 'marine ice cliff instability' was responsible for the Holocene retreat of Petermann Glacier from the GZW.

The paper is well written and easy to follow, with extensive reference to the relevant literature, and the dataset it analyses appears to be unique and valuable. The nature of the paper is rather long, heavy on description and in places very speculative. Whilst it appears to fit the stated aims and scope of the journal, I was surprised to see it here rather than in a longer-format journal such as JGR. Many concepts and processes are discussed, and I was left wondering whether this would not have been better divided into two papers – a general descriptive presentation of the data and overall glaciological insights; and a short paper focussing on the two significant issues identified above.

Nevertheless, I believe the paper worthy of publication, but have the following concerns and suggestions which need to be addressed or explained.

COMMENTS (in the order I noted them in the paper)

1) The manuscript relies heavily on the figures, which I chose to review on paper. They are generally well designed, but most of the annotation is too small and difficult to read. Double the size of fonts to get twice the engagement and communication.

We agree with the reviewer that the fonts may be too small in figures 2-6 if the paper is to be read on a printed version. We have therefore revised figures 2-6 and increased the font size substantially from 8 to 11 on all annotations.

2) Dates are given sometimes in calendar years and sometime in radiocarbon years, and reference is made to calibration more than once. This is not my area, but I feel that simplifying this situation would make the paper easier to understand.

We can certainly see that this may be confusing. However, we must mention the original dates in radiocarbon years of the ice retreat margins once, because this is the way England (1999) presented them. But we have moved up the part about how the calibration is made and placed it directly after where the ages are first mentioned, and then we use only calendar years from that point. We thus follow the reviewer's recommendation and removed the repetition of how they were calibrated to calendar years. We would like to keep the reference to the text in the caption of Figure 7 regarding that a calibration has been made since otherwise, readers who only look at the figure may be confused when comparing to England (1999).

Furthermore, we spotted an error on our behalf in that England (1999) had subtracted a reservoir age of 410 years to the radiocarbon years he presented. We have corrected for this

and added a sentence about it. We do believe there is no way we can avoid describing properly how the calibration from radiocarbon years to calendar years was made since it must be possible to re-produce by a reader, and we do also believe that we must present the ages in calendar years (as we now do), since radiocarbon years does not makes sense to the broader community and is practically not used anymore in final presentations of results.

The revised section reads:

“The ages of the ice retreat margins by England 36 will henceforth be provided in calendar years BP. Calibrations have been made on the ice retreat margins using the Marine13 radiocarbon age calibration curve 37 and a $\Delta R=268 \pm 82$ years, based on the five nearest analysed shells from Nares Strait and northern Baffin Bay 38. England 36 applied a reservoir correction of 410 years, which we accounted for when calibrating the radiocarbon years of the three ice margins presented in Figure 7. Furthermore, considering the uncertainty in assigning a representative ΔR for the Petermann Fjord area, calibrations of the ice margin ages using $\Delta R=0$ years are in addition presented in the caption of Figure 7.”

3) Likewise with water depths. Only in the Abstract is the discrepancy between present-day and palaeo water depths mentioned, yet this is crucial to some of the arguments. It should be made absolutely clear when depths are mentioned whether or not they are present day, and what impact the difference may have. Ideally, an explanation of the likely change in sea level since the formation of these features should be given, and water depths should be stated against the palaeo reference where that is relevant (e.g. MICI).

*We agree and we have clarified this in the revised manuscript. First, we add in the abstract (new text in red). “The **palaeodepth** limit for the recessional....” to make it absolutely clear that it is the deeper palaeodepth that supports MICI. Secondly, we added the following explanation to how the palaeodepth is estimated and the implications it has for MICI:*

*“The **present** depth of the Petermann Fjord (>1000 m), the retrograde landward slope of the GZW, and the location of iceberg plough ridges confirming calving all the way to the ice-stream base, lead us to suggest that MICI may have caused the Petermann Ice Stream to retreat rapidly from the fjord mouth sometime after 7600 cal. years BP. For ~1100 years before the rapid retreat, the ice stream was stable at the fjord mouth facilitated by frontal buttressing by an ice shelf, lateral pinning points, and a significant narrowing of the ice-stream trough. **The palaeodepth of the GZW and Petermann Fjord can be derived for a given time if the isostatic depression of the area is known. At 7000 cal. years BP, the isostatic depression has been estimated to ~70 m⁵²⁻⁵⁴, which implies that the point where MICI would occur is merely 50 m deeper than the deepest mapped morainic ridge on the retrograde landward slope of the GZW.**”*

4) The pseudo-3D figures do not present well on a 2D medium and are very difficult to understand. I recommend on paper using only plan-view figures.

This concerns figures 4 and 5, and we strongly believe that the 3D-views in this case provide a better illustration than a standard 2D map. While we agree that 3D-views may not always come out good on all printouts, primarily because the shadowing may get less distinct, we do believe that most readers will read the paper online and therefore hope we can maintain the 3D views.

5) The allusion to a “giant toilet seat” (a) shouldn't have double-quotes (or probably any

quotes at all) and (b) rather undermines the serious nature of the science. Use 'sliced donut' or something.

Quotes are removed in the text and figure captions. However, we decided to follow the advice of the reviewer and not using “toilet seats”, instead we resemble the mounds with “horseshoes”, which we find works very well as well.

6) In Figure 6, it appears that the iceberg prevented bathymetric acquisition resulting in edge effects in the shaded representation. This needs fixing with some kind of mask.

This is a correct observation. The iceberg did indeed prevent bathymetric acquisition underneath and we see an edge effect. In fact we see the grounded sidewalls of the iceberg as the multibeam captures them. This is clearly seen in a 3D view and since Reviewer 1 also raised doubts about whether or not the ploughmark is caused by this iceberg, we have added a small 3D view as an inset as it illustrates as well very well that this iceberg caused the iceberg ploughmarks.

7) Whilst possible, it is by no means certain (at least from the evidence presented) that the iceberg plough marks indicated were made by this iceberg, or even recently.

See above, we have made a small inset in Figure 6 that shows this clearly.

8) There is frequent reference to “stagnant (i.e. dead) ice”. The “i.e. dead” adds nothing to this – was ice ever “alive”?

Dead ice is in fact a commonly used term in glacial geology. However, we have changed “dead” for “stagnant” throughout the paper, except for where we mention “dead-ice moraines” once since it is a commonly used term and it does not work to say “stagnant-ice moraines” (which is not an adopted term).

9) The discussion of the origin of the donut features is so speculative as to undermine the whole paper, so I recommend you just state that the origin is unknown. Unless there is previous record of “giant moulins” or moulins “shaped like a horseshoe”, then the level of speculation is far too high.

We agree with Reviewer 1 that the formation mechanism for the horseshoe-shaped mounds is not central to the paper. However, Reviewer 2 brings up these as specifically interesting. We have decided to make a compromise. We include a description of the formation mechanism, but a more simplified and open ended version where the uncertainties are more clearly stated. We also suggest to remove figure 8, since it is not needed now when we highlight that there are parts of the formation that remains enigmatic, i.e. the central depression. The new text reads:

“Given the challenges of calling upon terrestrial analogues formed in stagnant ice for the formation of near circular mounds in a marine environment, we find that we cannot readily adopt any previously published formation mechanism for similarly looking mounds, e.g. kames⁴². However, we suggest a mechanism where sediment accumulates on stagnant ice in depressions, much like that commonly proposed for kames. The accumulated sediment must eventually make its way through the ice by meltwater aided transport, likely

through systems of moulins or crevasses, to be deposited in a confined high relief mound on the seafloor once the stagnant ice breaks up and disintegrates. The horseshoe-shape of some of the mounds remains an enigma. However, we speculate that it can potentially originate from meltwater that breaches the deposited sediment mound as it flows towards the deeper side of the seafloor.”

We have also added the following before this section:

Considering that the mounds are located next to mapped carbonate platforms of the lower Palaeozoic Washington Land Group formation on Kap Tyson^{39,41}, we cannot exclude that they may be carbonate mounds⁴². Against this interpretation is their well-preserved appearance, considering that they have been overridden by the Petermann Glacier, and that central depressions not are common features of carbonate mounds. Another possibility is that the mounds comprise a glacial landform.

10) The discussion of 'MICI' is confusing and needs more clarification. At one point the fjord depth (1000m) is used to invoke this argument, but line 446 seems to undermine this. These depths need better explanation with regard to palaeo sea level and isostasy. This is quite a big and trendy claim, so clarity on values is important.

We can see the confusion regarding palaeodepth and modern water depth and it may not have been clear how the isostasy came in. This has now been addressed, in the revised version, please see point 3 above.

11) The concluding paragraph is short and weak from such a lot of evidence. For people who tend to focus on the conclusions and figures, this paper would make little impression.

This paragraph has now been extended and more key points are included.

Reviewer #2 (Remarks to the Author):

This is a well written manuscript, although the detailed description of the morphological features is rather lengthy compared to the interpretation section and implication. On the other hand, this is the first detailed seafloor map of this section and it is an important contribution to understanding deglaciation in this part of Greenland. Some morphological features, especially the proposed formation of mounts resembling "toilet seats" through moulins is very interesting and innovative.

Detailed comments:

Lines 138-143: This introduction to this section could be skipped. That figures 4 and 5 are 3D views and figure 6 is based on RV Skidbladner is already stated in the figure captions.

We followed the recommendation and removed this section

Line 209 "small sinuous ridges". I can't really see them in fig 4a only in 4b.

This is one of the reasons for why we decided to include the slope maps, the features have very small bathymetric expression and are indeed very difficult to see in a bathymetric shaded relief. .

Line 215/216: recessional moraine have been reported from tide water glaciers, but it is not clear to me why they couldn't be formed by glaciers with an ice shelf. Tide-related movement of the grounding line has been reported for ice streams with ice selves in Antarctica (e.g. Bindshaler et al 2003 (10.1126/science.1087231), although we don't know, if they produce such ridges.

We also assume that small ridges can be created at the grounding line by glaciers with an ice shelf. But the pattern we see shows successive back stepping, and then it is hard to envision how that this would happen if an ice shelf is in place, since the mass must then be lost very regularly and in increments from melting. We have clarified this in the paper by adding:

The pattern of the ridges suggests a regular back-stepping of the glacier margin, which makes us suggest that there was no ice tongue in place when they were formed.

Line 333: "We have no morphological data". I suggest dropping "morphological since there are probably no other data, e.g. from sediments, either. Ice shelf presence is usually inferred from sediments rather from seafloor morphology.

Revised as suggested

Line 412 switch "Figs 2cb" to "Figs 2 b, c"

Revised as suggested

Figures

Figure 1:

- I suggest adding a label for Nare Strait on 1b (There is a label NS in 1a, but is not explained in captions)

Both revised as suggested

- while the figures are very well drafted, the coloring of the labels (e.g. white on light blue) or black on dark blue is sometimes hard to see.

We have changed several of the labels to black where the contrast with white became too low. On request of Reviewer 1 we also increased the font size which helps making all labels more legible.

- maybe mark the older ice tongue locations as dashed white lines instead of black lines

Revised as suggested

- consider changing labels S1-S4 to darker color, e.g. black.

Revised as suggested

Figure 2

I am not sure the slope maps 2b, d, f, are necessary. In most cases the glacial features are clearer in the bathymetry maps with hill-shading. I suggest moving the slope maps to the supplement material.

If there is no editorial reason for moving them to supplementary material we really prefer to keep them, specifically since the recessional small moraines are so clear in the slope maps, and we fear that most readers to not go to the supplementary material.

Where the bathymetry is shown in light blue (2c, e) the labels would be easier to read in a darker color.

This has been changed, where the contrast is better in black, the labels are changed.

Figure 3:

Similar to Fig 2 I am not sure the slope maps 3b, d are necessary. In most cases the glacial features are clearer in the bathymetry maps with hill-shading. I suggest moving the slope maps to the supplement material.

For the same reason as above, we prefer to keep them. We do believe they provide added value in the main article.

Where the bathymetry is shown in light blue (3c) the labels would be easier to read in a darker color.

Revised as suggested

Figure 4:

Add a view direction to 3D view in Fig 4a and b

View directions have been added for the 3D views.

Figure 5:

Add a view direction to 3D view in Fig 4a and orientation to 5b

View directions have been added for the 3D views. We are not shore what is meant by orientation, but with the new north arrow in place is it rather clear how the profile is oriented.

Figure 7:

There is yellow-blue-green circle in the top right corner. I don't think this marks an ice position but might have been a legend for the ice extent markers without labels. I suggest removing it or and labels.

Labels are added since it a local ice field according to England (1999).

It would be good to add the color indicator after the years in line 300. E.g. "8700 (green), 8150 (yellow), 7200 (blue)".

Revised as suggested